# Presence of antiphospholipid antibodies is associated with increased implantation failure following *in vitro* fertilization technique and embryo transfer: A systematic review and meta-analysis

Eirini Papadimitriou[1,2], Georgios Boutzios[1], Alexander G. Mathioudakis[3,4], Nikos F. Vlahos[5], Panayiotis Vlachoyiannopoulos[1], George Mastorakos[2]*

1 Department of Pathophysiology, Laiko University Hospital, Medical School, National and Kapodistrian University of Athens, Athens, Greece, 2 Endocrine Unit, Aretaieion University Hospital, Medical School, National and Kapodistrian University of Athens, Athens, Greece, 3 Division of Infection, Immunity and Respiratory Medicine, School of Biomedical Sciences, The University of Manchester, Manchester, United Kingdom, 4 North West Lung Centre, Wythenshawe Hospital, Manchester University NHS Foundation Trust, Manchester Academic Health Science Centre, Manchester, United Kingdom, 5 2nd Department of Obstetrics and Gynecology, Aretaieion University Hospital, Medical School, National and Kapodistrian University of Athens, Athens, Greece

* mastorakg@gmail.com

## Abstract

### Purpose

A systematic review and meta-analysis was conducted comparing the presence of anti-phospholipid (anti-PL) antibodies between women of reproductive age, without diagnosis of antiphospholipid syndrome, who experienced at least two implantation failures following *in vitro* fertilization and embryo transfer (IVF-ET), and either women who had a successful implantation after IVF-ET or women with at least one successful spontaneous pregnancy or unselected healthy fertile women with no history of IVF-ET.

### Methods

Systematic search of the literature and meta-analysis of the relevant studies studying presence of antiphospholipid antibodies in women experiencing at least two implantation failures in IVF-ET as compared to either women who had a successful implantation after IVF-ET or/ and women with at least one successful spontaneous pregnancy or unselected healthy fertile women with no history of IVF-ET. Six hundred ninety-four published reports were retrieved; 17 of them fulfilled the inclusion criteria set.

### Results

Presence of either any type of anti-phospholipid or anticardiolipin antibodies or lupus-anticoagulant in women experiencing at least two implantation failures in IVF-ET was associated with increased implantation failure compared to women who had a successful implantation

**Data Availability Statement:** This is a systematic review and meta-analysis of published data that are referenced in the manuscript and no new data was collected as part of this work.

**Funding:** The authors received no specific funding for this work.

**Competing interests:** The authors have declared that no competing interests exist.

after IVF-ET (relative risk, RR: 3.06, 5.06 and 5.81, respectively). Presence of either anticardiolipin or lupus-anticoagulant or anti-beta$_2$ glycoprotein-I or anti-phosphatidylserine antibodies in women experiencing at least two implantation failures in IVF-ET was associated with increased implantation failure compared to unselected healthy fertile women with no history of IVF-ET (RR:13.92, 6.37, 15.04 and 164.58, respectively).

## Conclusion

The prevalence of antiphospholipid antibodies, particularly that of anti-beta$_2$ glycoprotein-I and anti-phosphatidylserine antibodies, in women experiencing at least two implantation failures in IVF-ET without diagnosis of antiphospholipid syndrome is significantly greater than either in women who had a successful implantation after IVF-ET or women with at least one successful spontaneous pregnancy or unselected healthy fertile women with no history of IVF-ET.

## Trial registration number

PROSPERO ID: CRD42018081458

## Introduction

Infertility is a public health problem which affects 1:10 women of reproductive age [1, 2]. Its estimated world prevalence is 186 million people [1, 2]. Assisted reproductive technology (ART) has led to a significant rise in live births following the introduction of *in vitro* fertilization (IVF) [3]. Over 1,250,000 ART cycles, resulting in birth of over 225,000 babies, were reported by 2,419 clinics globally in 2007. The availability of ART varies by country, from 12 to 4,140 treatments per million population [4].

Rheumatic diseases can affect quality of life and reproduction. Pregnancy complications are increased in patients with systemic lupus erythematosus and antiphospholipid syndrome (APS). The latter is an autoimmune acquired thrombophilia, which occurs either alone or in combination with other autoimmune diseases, mainly with systemic lupus erythematosus [4]. Antiphospholipid antibodies represent a heterogeneous group of antibodies, which recognize various phospholipids, phospholipid-binding proteins, and phospholipid protein complexes. Clinical manifestations of APS include fertility problems and pregnancy complications (such as repeated miscarriages) as well as venous or arterial thrombosis [5]. Evaluation of circulating anti-phospholipid (anti-PL) antibodies is part of the serological work-up following miscarriage. When circulating anti-PL antibodies are positive at initial diagnosis, testing should be repeated at least 12 weeks later to confirm diagnosis of APS [5]. According to revised Sapporo criteria, diagnosis of APS takes into account lupus anticoagulant (LA), anti-cardiolipin (anti-CL) antibodies or anti-$\beta_2$glycoprotein I (anti-$\beta_2$GP I) antibodies of either IgG or IgM isotype.

The relationship between presence of anti-PL antibodies (without diagnosis of APS) and implantation failure has been examined by several original studies which suggested that presence of anti-PL antibodies, even without diagnosis of APS, impairs implantation. Antiphospholipid antibodies, especially anti-beta$_2$ glycoprotein I (anti-$\beta_2$GPI) antibodies, in pregnancy, appear to act directly on trophoblasts by activating pro-apoptotic and pro-inflammatory mechanisms [6]. At the same time, thrombosis of placental chorionic arteries and

activation of the complement system intravascularly lead to the cell death of the trophoblast by decreasing trophoblast viability, syncytialization, and capacity for invasion [6].

Whether the presence alone of anti-PL antibodies in healthy women of reproductive age who do not fulfill the criteria for APS, might affect implantation and embryo transfer (ET) following IVF, is not decided as yet in the literature. To fill this gap this systematic review and meta-analysis were conducted.

## Material and methods

### Protocol

**Search strategy and selection of studies.** This systematic review and meta-analysis was based on a protocol registered prospectively in PROSPERO database for systematic review protocols (ID: CRD42018081458) and follows Preferred reporting Items for Systematic Reviews and Meta-Analyses (PRISMA) statement [7, 8]. The electronic databases of Medline (Pubmed) and Cochrane library were reviewed systematically from inception to April 2021, using appropriate controlled vocabulary and free search terms to identify studies evaluating fertility in women in association with presence of any type of anti-PL antibodies (detailed search strategy is available in the online Appendix 1 in S1 Appendix). Titles, abstracts and full text (when appropriate) of all identified studies were screened for eligibility by one author (E.P.). The same author extracted from the studies the following pieces of information in a pre-specified standardized MS Excel: full reference; study identifiers; study design; eligibility; predefined outcomes; number of participants (population index and controls); characteristics of participants; details on the outcomes of interest. The term *population index* refers to the total number of women who experienced at least two implantation failures after IVF-ET. Search strategy was validated by GM. When EP raised a discrepancy, GM was consulted. The extracted characteristics of participants were: age; cause of subfertility wherever applicable; number of years of subfertility wherever applicable; numbers of IVF/ET attempts wherever applicable; number of retrieved and fertilized oocytes; quality of embryos (defined as regular blastomeres, or according to the presence of even cleavage; even cell sizes; less than 20% fragmented blastomeres); number of transferred embryos wherever applicable; past medical history of women included in each study was retrieved (no women suffered from APS or had a history of thrombosis); time period in which participants were enrolled; provenance of participants; laboratory technique for measurement of any type of anti-PL antibodies. All these steps were validated by a second reviewer (A.G.M.). Disagreement was resolved by discussion or adjudication by a third investigator if necessary (G.M.)

**Criteria for inclusion of studies in the meta-analysis.** Studies fulfilling all of the following criteria were included in this meta-analysis:

i. Studies published in English with prospective or retrospective observational design.

ii. All study populations should be consisted by healthy women of reproductive age not suffering from any known autoimmune, endocrine or infectious diseases.

iii. Studies comparing the prevalence of any type of anti-PL antibodies between women experiencing at least two implantation failures in IVF-ET (population index) *vs.* either women experiencing one successful IVF-ET or women with at least one successful spontaneous pregnancy or unselected healthy fertile women with no history of IVF-ET. Studies with control women experiencing one successful IVF-ET were included in *subgroup A of selected studies* [9–14], while studies with control women with at least one successful spontaneous pregnancy or unselected healthy fertile women with no history of IVF-ET were included in *subgroup B of selected studies* [3, 9, 13, 15–24].

**Study outcomes extracted for the meta-analysis.** The primary outcome extracted from the selected studies was presence or not of any type of anti-PL antibodies. Secondary outcomes extracted from the selected studies were presence or not of: anticardiolipin (anti-CL), lupus anticoagulant (LA) and anti-β2GPI antibodies (all three representing aPL included in the Sapporo criteria for APS diagnosis), as well as anti-phosphatidylserine (anti-PS), anti-phosphatidylcholine (anti-PC), anti-phosphatidylethanolamin (anti-PE), anti-phosphatidylinositol (anti-PI), anti-phosphatidylglycerol (anti-PG) and anti-phosphatidic acid (anti-PA) antibodies which have gained importance in recent literature for APS diagnosis.

## Risk of bias assessment

Risk of bias was assessed by two authors independently (E.P. and A.G.M.) by employing the Newcastle-Ottawa Scale [25]. Disagreement was resolved by discussion or adjudication by a third investigator if necessary (G.M.). In line with the previously submitted protocol of the meta-analysis, a publication bias analysis (Funnel plot and Egger's) was not performed because comparisons consisted of less than 25 eligible studies render such analysis less informative [26].

## Statistical analyses

Heterogeneity among selected studies was evaluated in each analysis, using $I^2$ statistic [26, 27]. According to Cochrane handbook: $I^2 \geq 75\%$, $I^2$ between 75% and 50% or $I^2 \leq 50\%$ reflect substantial, significant or non-significant heterogeneity among the selected studies, respectively [27]. When $I^2$ was $\geq 75\%$, possible causes of heterogeneity were investigated by performing pre-specified subgroup analyses. Meta-analysis was not performed in case of substantial heterogeneity which could not be resolved by subgroup analysis. In these cases, findings were reported narratively.

All outcomes were dichotomous and were analyzed by calculating relative risks (RR) and 95% confidence intervals (CI). For data synthesis the random effect model was employed as significant clinical and methodological heterogeneity among the included studies was anticipated. In addition, *a priori* specified subgroup analyses were conducted to explore significant or substantial heterogeneity and to further evaluate the soundness of results.

In a pre-specified sensitivity analysis, meta-analyses were repeated using the *fixed effects model*. All analyses were performed using Review Manager 5.3 Software [28].

**Ethical approval.** This article does not contain any studies with human participants or animals performed by any of the authors.

## Results

### Search strategy for the systematic review

Search strategy identified 621 and 73 references in Medline and Cochrane Library, respectively (Fig 1). After removal of duplicate records between the two databases, a total of 629 studies were screened by title and abstract. Of these references, 40 were deemed potentially eligible because they reported studies regarding healthy women of reproductive age with implantation failure after IVF-ET and were assessed by full-text. Twenty-three of these 40 studies were excluded because either the outcome was not precisely reported or their design did not fulfill the inclusion criteria set. In the remaining 17 studies included in the present systematic review and meta-analysis (ten and seven studies were retrospective and prospective cohort studies, respectively), 4,075 healthy women of reproductive age were evaluated. Selection process is

**PRISMA 2020 flow diagram**

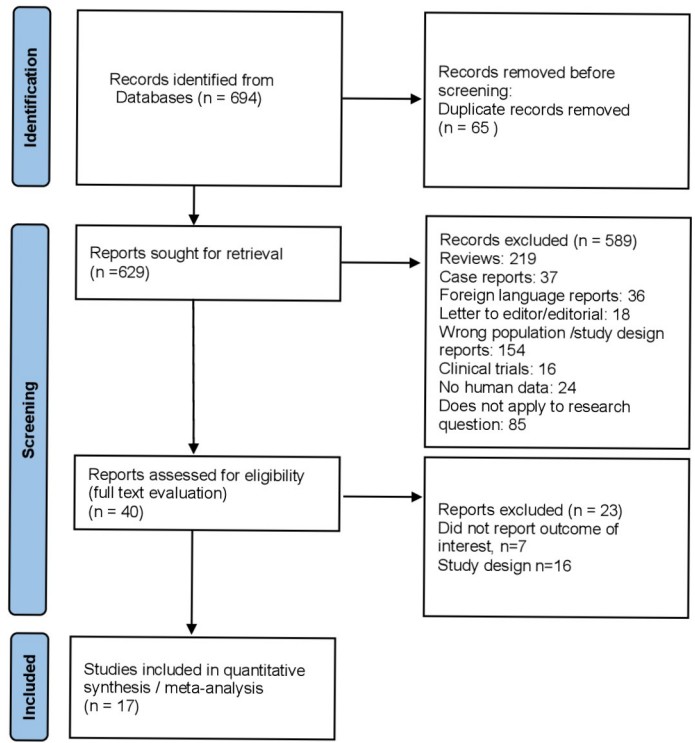

**Fig 1. PRISMA flow diagram of the selection process from identified studies to selected studies through Medline and Cochrane Library.**

described in Fig 1, while the characteristics of each of the included studies are reported in the online Appendix 2 in S1 Appendix.

## Characteristics of the women in the selected studies

The women with implantation failure [absence of positive pregnancy tests based on beta human chorionic gonadotropin (hCG) evaluation; in none of the studies the timing of hCG measurement was reported] after IVF-ET included in all studies had at least two implantation failures (range of failures: 2–6) following transfer of good quality embryos. Three studies included confirmation of a gestational sac by ultrasound three weeks after embryo transfer [9, 16, 18]. All women included in these studies were not receiving any additional treatment.

**Subgroup A of selected studies.** Studies of this subgroup, included women referred for IVF-ET due to similar indications [9–14]. Unexplained infertility was the predominant indication, followed by tubal factor -for the majority of them. Age, duration and type of infertility did not differ between population index and controls studied in this subgroup.

**Subgroup B of selected studies.** In these studies, the majority of patients received IVF for unexplained infertility. In studies of this *subgroup*, other, not frequently encountered, indications for IVF-ET were endometriosis, ovulation disorders or mixed infertility.

## Characteristics of the included studies in the systematic review

**Ovarian stimulation protocols employed.**   Four studies reported that a standard protocol for ovarian stimulation was followed (a combined regimen of GnRH agonists and human menopausal gonadotrophins) [9–11, 18]. The remaining studies did not report the specific ovarian stimulation protocol employed.

**Assays employed for the antibodies evaluation.**   All selected studies employed enzyme-linked immunosorbent assays (ELISA) for presence of any type of anti-PL antibodies. Lupus anticoagulant was evaluated either by the kaolin cephalin clotting time utilizing sensitive reagents or by the dilute Russell's viper venom time with a neutralization procedure using frozen–thawed platelets or by both techniques. Results for anti-PL and LA antibodies were expressed as positive or negative. Diagnostic cut-offs for the antibodies were reported in: Qublan H. *et al.* (anti-CL antibodies positive >10 IU/ml; qualitative positivity or negativity for LA) [13]; Vaquero E. *et al.* (anti-CL antibodies positive >20; qualitative positivity or negativity for LA) [18]; Bellver *et al.* (anti-CL antibodies positive >20 gPL/ml or mPL/ml for IgG or IgM isotype, respectively; qualitative positivity or negativity for LA) [21]; Sanmarco M. *et al.* [antibodies positive for anti-CL: IgG ≥20 GPLU; for anti-β2GPI IgG ≥10 B2GU; for aPE IgG ≥15 PEGU (GPLU, MPLU, B2GU and PEGU are arbitrary units for optical density)] [20]; Geva E. *et al.* (anti-CL antibodies positive >23 GPLU) [12].

In six studies positivity was based on optical density measurement exceeding the 99th or the 95th percentile of measurements established for each phospholipid in healthy individuals of reproductive age [11, 14, 17, 19, 22, 23]. Six studies did not report diagnostic cut-offs [3, 9, 10, 15, 16, 24]. Manufacturers of the assays are reported in all studies except two [13, 20].

**Subgroup A of selected studies.**   Six studies (n = number of women studied; n = 438) evaluated presence of any type of anti-PL antibodies (anti-CL, anti-PS, anti-PC, anti-PE, anti-PI and anti-β$_2$GPI antibodies) [3, 9–13]. Four studies (n = 360) evaluated presence of anti-CL antibodies [9, 11–13]. Two studies (n = 268) evaluated presence of LA [9, 13]. One study (n = 42) evaluated presence of anti-PS, anti-P, anti-PC, anti-PE, anti-PA and anti-PG antibodies [13]. Meta-analysis was performed for the prevalence of either any type of anti-PL antibodies or for anti-CL antibodies or for LA antibodies.

**Subgroup B of selected studies.**   Thirteen studies (n = 3,637) evaluated the presence of any type of anti-PL antibodies [3, 9, 13, 16–24].

Six studies (n = 2610) evaluated the presence of anti-CL antibodies [9, 12, 14, 18, 20, 21]. Three studies (n = 2004) evaluated the presence of anti-CL-IgG as well as anti-CL-IgM antibodies [18, 20, 21]. Four studies (n = 353) evaluated the presence of LA antibodies [9, 12, 15, 20]. Three studies (n = 2144) evaluated the presence of anti-β$_2$GP [9, 18, 22]. Two studies (n = 1978) evaluated the presence of anti-PS antibodies [18, 21]. One study (n = 1926) evaluated the presence of anti-PI, anti-PA, anti-PE and anti-PG antibodies [18]. Meta-analysis was performed for the prevalence of either anti-CL-IgG or LA or anti-β$_2$GPI or anti-PS antibodies.

Qublan et al. and Khizroeva et al. evaluated women with implantation failures after IVF-ET compared with both control groups. Therefore, these studies are included in both subgroups A and B of selected studies [9, 13].

Data regarding antibodies studied only in one study in each subgroup were not included in a meta-analysis. Thus, neither data on anti-P, anti-PC, anti-PE, anti-PA, anti-PG and anti-PS antibodies reported only in the study by Kaider et al. (Subgroup A) nor data on anti-P, anti-PE, anti-PA and anti-PG antibodies reported only in the study by Ulcova-Gallova et al. (Subgroup B) were included in a meta-analysis (Tables 1 and 2).

**Table 1. Prevalence (reported as percentages in parentheses) of different types of anti-phospholipid (anti-PL) antibodies in studies included in subgroup A of studies.**

|  | any type of anti-PL | Anti-CL | LA | anti-P | anti-PC | anti-PE | anti-PA | anti-PG | anti-PS |
|---|---|---|---|---|---|---|---|---|---|
| *Birkenfeld et al.* | 18/56 (32.1%) *vs* 0/14 (0%) |  |  |  |  |  |  |  |  |
| *Buckingham et al.* | 5/22 (22.7%) *vs* 13/71(18.3%) |  |  |  |  |  |  |  |  |
| *Geva et al.* | 3/50 (6%) *vs* 0/40 (0%) | 3/50 (6%) *vs* 0/40 (0%) |  |  |  |  |  |  |  |
| *Qublan et al.* | 17/90 (18.9%) *vs* 4/90 (4.4%) | 9/90 (10%) *vs* 2/90 (2.2%) | 8/90 (8.9%) *vs* 2/90 (2.2%) |  |  |  |  |  |  |
| Khizroeva *et al.* | 75/178(42.1%) *vs* 22/169 (13%) | 16/178 (9%) *vs* 3/169 (1.8%) | 35/178 (19.7%) *vs* 5/169 (3%) |  |  |  |  |  |  |
| *Kaider et al.* | 11/42 (26.2%) *vs* 0/42 (0%) | 3/42 (7.1%) *vs* 0/42(0%) |  | 3/42 (7.1%) *vs* 0/42(0%) | 12/42 (21.4%) *vs* 0/42(0%) | 3/42 (7.1%) *vs* 2/42(4.8%) | 3/42(7.1%) *vs* 0/42(0%) | 2/42(4.8%) *vs* 0/42(0%) | 1/42 (2.4%) *vs* 0/42(0%) |

*Footnote*: Studies in subgroup A compare women with at least two implantation failures in IVF-ET vs. women with one successful IVF-ET. Anti-PL: anti-phospholipid antibodies; anti-CL: anti-cardiolipin, antibodies; LA: lupus anticoagulant; anti-PI: anti-phosphatidylinositol antibodies; anti-PC: anti-phosphatidylcholine antibodies; anti-PE: anti-phosphatidylethanolamin antibodies; anti-PA: anti-phosphatidic acid antibodies; anti-PG: anti-phosphatidylglycerol antibodies and anti-PS: antiphospatidilserine antibodies.

## Risk of bias assessment

Based on the Newcastle Ottawa scale all studies of subgroup A were rated as of low risk in all assessed domains: selection bias of population index and control, performance bias, detection bias and attrition bias.

Ten out of thirteen studies of subgroup B (77%) were rated as having low risk and three (23%) as having unclear risk of bias regarding selection of population index. Regarding selection of controls, five out of thirteen studies of subgroup B (39%) were rated with unclear risk of bias. All studies of subgroup B were rated with low risk of bias regarding performance bias, detection bias and attrition bias. Detailed assessment of the risk of bias is available in online Appendix 3 in S1 Appendix.

## Outcomes of systematic review and meta-analysis

The reported results are the outcome of separate random effect analyses of *subgroup A* and *subgroup B*. The sensitivity analyses using fixed effect methods showed similar results.

The majority of selected studies supported the association between presence of any type of anti-PL antibodies and infertility. Studies included in subgroup A were fairly homogenous ($I^2$ range: 0–15%) and provided quite reliable results, whereas studies included in subgroup B were less homogeneous.

**Outcomes of subgroup A of selected studies (Table 1).** Twenty-nine percent and 9.6% of population index and controls respectively, reported presence of any type of anti-PL antibodies among anti-CL, anti-$\beta_2$GPI, anti-PS, anti-PC, anti-PE, anti-PI, anti-PA and anti-PG antibodies of IgG, IgM or IgA isotypes (six studies) [3, 9–13]. Population index showed a RR for the presence of any type of the above mentioned anti-PL antibodies of 3.06 for implantation failure (95% CI: 1.97, 4.77, $I^2$ = 15%) compared to controls (Fig 2). Because the anti-PC and anti-PE antibodies have lost part of their importance in the recent literature, a meta-analysis has been also performed without including the only study which evaluated these antiphospholipid antibodies. The obtained RR did not change substantially [RR: 2.89 (95% CI: 1.73, 4.81), $I^2$ = 22%]. In addition, 8.6% and 1.5% of population index and controls, respectively,

**Table 2. Prevalence (reported as percentages in parentheses) of different types of anti-phospholipid (anti-PL) antibodies in studies included in subgroup B of studies.**

| | any type of anti-PL | anti-CL | anti-CL Ig-G | anti-CL Ig-M | LA | anti-β$_2$GPI | anti-PI | anti-PE | anti-PA | anti-PG | anti-PS |
|---|---|---|---|---|---|---|---|---|---|---|---|
| *Alves et al.* | 48/52 (92%) vs 0/28 (0%) | 50/52 (96.2%) vs 0/28 (0%) | 29/52 (55.8) vs 0/28 (0%) | 50/52 (96%) vs 0/28 (0%) | | | | | | | 48/52 (92.3) vs 0/42 (0%) |
| *Bellver et al.* | 8/57 (14%) vs 6/32 (18.8%) | 8/57 (14%) vs 6/32 (18.8%) | 1/26 (3.8%) vs 0/32 (0%) | 1/26 (3.8%) vs 6/32 (18.8%) | 3/26 (11.5%) vs 0/32 (0%) | | | | | | |
| *Coulam 97 et al.* | 69/312 (22%) vs 5/100 (5%) | 13/312 (4.2%) vs 0/100 (0%) | | | | | | | | | |
| *Coulam 02 et al.* | 34/122 (27.9%) vs 7/107 (6.5%) | | | | | | | | | | |
| *Paulmyer-Lacroix et al.* | 8/40 (20%) vs 1/100 (1%) | | | | | 5/40 (12.5%) vs 1/100 (1%) | | | | | |
| *Qublan et al.* | 17/90 (18.9%) vs 9/100 (9%) | 9/90 (10%) vs 3/100 (3%) | | | 8/90 (8.9%) vs 2/100 (2%) | | | | | | |
| *Sanmarco et al.* | 40/101 (39.6%) vs 8/160 (5%) | | | | | | | | | | |
| *Steinvil et al.* | 17/509 (3.3%) vs 30/637 (4.7%) | | | | | | | | | | |
| *Stern et al.* | 30/105 (28.6%) vs 16/106 (15.1%) | | | | | | | | | | |
| *Ulcova-Gallova et al.* | 928/1926 (48.2%) vs 5/391 (1.3%) | 421/1926 (21.9%) vs 8/391 (2%) | 349/1926 (18.6%) vs 5/391 (1.3%) | 72/1926 (3.7%) vs 3/391 (0.8%) | | 209/1926 (10.9%) vs 0/391 (0%) | 613/1926 (31.8%) vs 0/391 (0%) | 377/1926 (19.6%) vs 2/391(0.5%) | 240/1926 (12.5%) vs 0/391 (0%) | 318/1926 (16.5%) vs 0/391 (0%) | 778/1926 (40.4%) vs 0/391 (0%) |
| *Vaquero et al.* | 8/59 (13.6%) vs 0/20 (0%) | | | | 3/59 (5%) vs 0/20 (0%) | | | | | | |
| *Khizroeva et al.* | 75/178 (42.1%) vs 3/80 (3.8%) | 16/178 (9%) vs 1/80 (1.3%) | | | 35/178 (19.7%) vs 1/80 (1.3%) | 56/178 (31.5%) vs 3/80 (3.8%) | | | | | |
| *Saxtorph et al. 2020* | 2/86 (2%) vs 0/37 (0%) | | | | | | | | | | |

*Footnote*: Subgroup B compares women with at least two implantation failures in IVF-ET vs. either women with at least one successful spontaneous pregnancy or unselected healthy fertile women with no history of IVF-ET. Anti-PL: anti-phospholipid antibodies; anti-CL: anti-cardiolipin, antibodies; LA: lupus anticoagulant; anti-β$_2$GPI: anti-β$_2$glycoprotein I antibodies; anti-PI: anti-phosphatidylinositol antibodies; anti-PE: anti-phosphatidylethanolamin antibodies; anti-PA: anti-phosphatidic acid antibodies; anti-PG: anti-phosphatidylglycerol antibodies and anti-PS: anti-phospatidilserine antibodies.

had present anti-CL antibodies (four studies) [9, 11–13]. Population index showed a RR for the presence of anti-CL antibodies of 5.06 for implantation failure (95% CI: 2.14, 11.95; I$^2$ = 0%) compared to controls (Fig 3). Moreover, 16% and 2.7% of population index and controls, respectively, had LA antibodies present (two studies) [9, 12]. Population index showed a RR

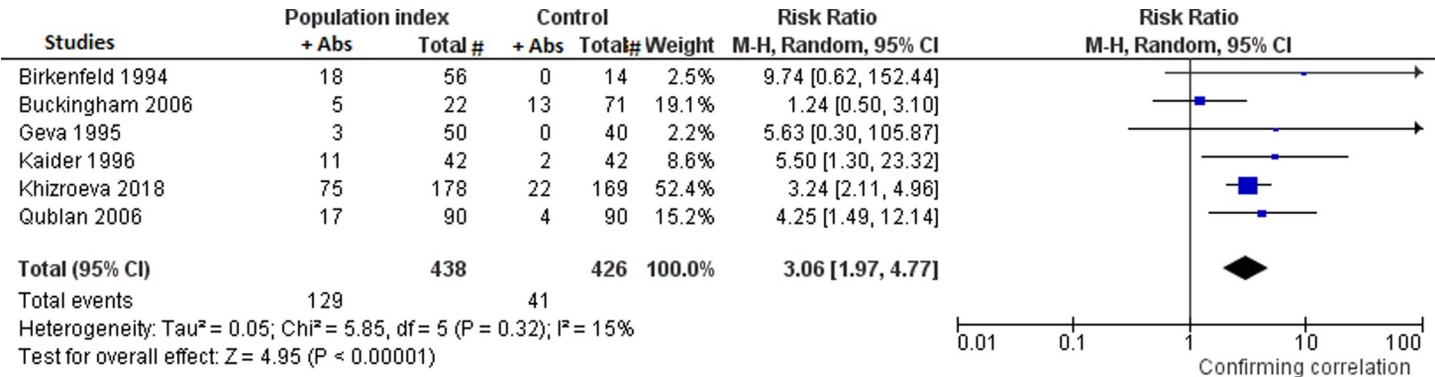

**Fig 2. Meta-analyses assessing the risk for implantation failure in relation to the presence or not of anti-PL antibodies in the studies in subgroup A between women with at least two implantation failures in IVF-ET (*population index*) *vs.* women with one successful IVF-ET (*control*).** Fig 2: meta-analysis assessing the risk for implantation failure in relation to the presence or not of any type of anti-PL antibodies.

for presence of LA antibodies of 5.81 for implantation failure (95% CI: 2.66, 12.71; $I^2 = 0\%$) compared to controls (Fig 4).

**Outcomes of subgroup B of selected studies (Table 2).** Three studies out of six in subgroup B point out two distinct aCL isotypes (aCL/Ig-G, aCL/Ig-M) in their reporting of aCL evaluation. However, meta-analysis was conducted only for the aCL/Ig-G isotype because, for this isotype, a non-significant heterogeneity among the selected studies was observed, whereas for the aCL/Ig-M isotype the heterogeneity observed was substantial. Thus, in line with the methodology and guidance of Cochrane handbook, a meta-analysis was not meaningful for the aCL/Ig-M isotype. Nineteen percent and 1% of population index and controls, respectively had anti-CL-IgG antibodies present (three studies) [18, 20, 21]. Presence of anti-CL antibodies of the IgG isotype was more strongly associated with implantation failure after IVF-ET compared to that of the IgM isotype. Specifically, population index showed a RR for the presence anti-CL antibodies of the IgG isotype of 13.92 for implantation failure (95%CI: 6.21, 31.21; $I^2 = 0\%$) compared to controls (Fig 5).

Fourteen percent and 1.3% of population index and controls, respectively, had LA antibodies present (four studies) [9, 12, 15, 20]. Population index showed a RR for the presence of LA antibodies of 6.37 for implantation failure (95% CI: 2.25, 18.04; $^2 = 0\%$) compared to controls (Fig 6).

Almost thirteen percent and 0.7% of population index and controls, respectively, had anti-$\beta_2$GPI antibodies present (three studies) [9, 18, 22]. Population index showed a RR for the

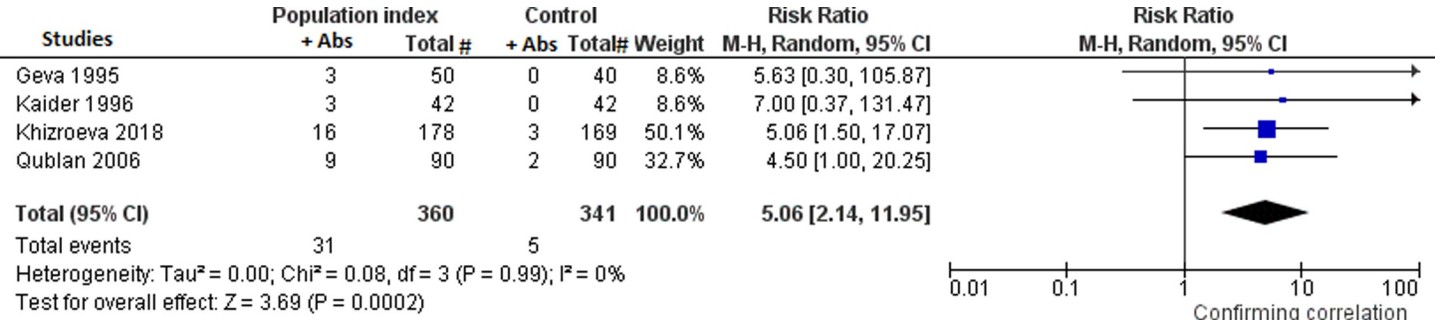

**Fig 3. Meta-analyses assessing the risk for implantation failure in relation to the presence or not of anti-PL antibodies in the studies in subgroup A between women with at least two implantation failures in IVF-ET (*population index*) *vs.* women with one successful IVF-ET (*control*).** Fig 3: meta-analysis assessing the risk for implantation failure in relation to the presence or not of anti-CL antibodies.

| Studies | Population index | | Control | | | Risk Ratio | Risk Ratio |
| | + Abs | Total# | + Abs | Total# | Weight | M-H, Random, 95% CI | M-H, Random, 95% CI |
|---|---|---|---|---|---|---|---|
| Khizroeva 2018 | 35 | 178 | 5 | 169 | 73.5% | 6.65 [2.67, 16.56] | |
| Qublan 2006 | 8 | 90 | 2 | 90 | 26.5% | 4.00 [0.87, 18.32] | |
| Total (95% CI) | | 268 | | 259 | 100.0% | 5.81 [2.66, 12.71] | |
| Total events | 43 | | 7 | | | | |
| Heterogeneity: Tau² = 0.00; Chi² = 0.32, df = 1 (P = 0.57); I² = 0% | | | | | | | |
| Test for overall effect: Z = 4.40 (P < 0.0001) | | | | | | | |

**Fig 4. Meta-analyses assessing the risk for implantation failure in relation to the presence or not of anti-PL antibodies in the studies in subgroup A between women with at least two implantation failures in IVF-ET (*population index*) *vs*. women with one successful IVF-ET (*control*).** Fig 4: meta-analysis assessing the risk for implantation failure in relation to the presence or not of LA antibodies. +Abs: positive antibodies, total #: total number of participants.

presence of anti-$\beta_2$GPI of 15.04 for implantation failure (95% CI: 3.47, 65.10; $^2$ = 44%) compared to controls (Fig 7).

Two studies assessed anti-PS antibodies [18, 21]. Forty-two percent of population index had anti-PS antibodies present, as compared 0% of controls. Population index showed a RR for the presence of anti-PS of 164.58 for implantation failure (95% CI: 23.31, 1162.26; $I^2$ = 0%), compared to controls (Fig 8).

Two studies of this subgroup did not show any association between the presence of any type of anti-PL or anti-CL antibodies and IVF-ET outcome [3, 21]. However, the results of both studies should be examined with caution as *Bellver et al.* were based on a very limited study population, while it is unclear how *Steinvil et al.* selected their control group.

## Discussion

Among 629 references that this systematic search yielded from Medline and Cochrane Library, a limited number of 17 studies, involving 4,075 women of reproductive age, were included in this systematic review and meta-analysis. The addition of good studies in the future will improve the accuracy of the deduced conclusions. All included studies involved women with at least two implantation failures in IVF-ET *vs.* either women with one successful IVF-ET or women with at least one successful spontaneous pregnancy or unselected healthy fertile women with no history of IVF-ET. When a specific type of anti-PL antibodies was evaluated in heterogeneous or single studies, the reported data were not included in the meta-analysis.

We found, in this meta-analysis, that in women experiencing at least two implantation failures in IVF-ET, presence of either any type of anti-PL antibodies or anti-CL antibodies only or LA antibodies is associated with a significant 3.06, 5.06 and 5.81 RR for impaired implantation

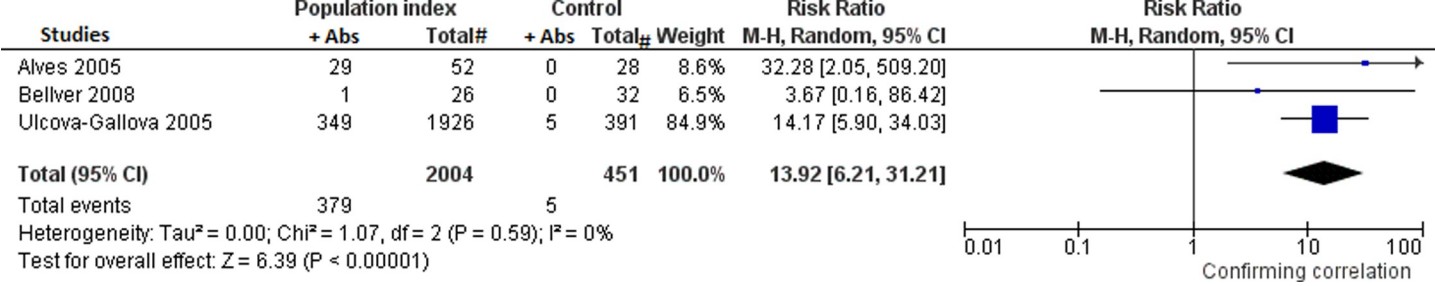

| Studies | Population index | | Control | | | Risk Ratio | Risk Ratio |
| | + Abs | Total# | + Abs | Total# | Weight | M-H, Random, 95% CI | M-H, Random, 95% CI |
|---|---|---|---|---|---|---|---|
| Alves 2005 | 29 | 52 | 0 | 28 | 8.6% | 32.28 [2.05, 509.20] | |
| Bellver 2008 | 1 | 26 | 0 | 32 | 6.5% | 3.67 [0.16, 86.42] | |
| Ulcova-Gallova 2005 | 349 | 1926 | 5 | 391 | 84.9% | 14.17 [5.90, 34.03] | |
| Total (95% CI) | | 2004 | | 451 | 100.0% | 13.92 [6.21, 31.21] | |
| Total events | 379 | | 5 | | | | |
| Heterogeneity: Tau² = 0.00; Chi² = 1.07, df = 2 (P = 0.59); I² = 0% | | | | | | | |
| Test for overall effect: Z = 6.39 (P < 0.00001) | | | | | | | |

**Fig 5. Meta-analysis assessing the risk for implantation failure in relation to the presence or not of anti-PL antibodies in the studies in subgroup B between women with at least two implantation failures in IVF-ET (*population index*) *vs* women with at least one successful spontaneous pregnancy or unselected healthy fertile women with no history of IVF-ET (*control*).** Fig 5: meta-analysis assessing the risk for implantation failure in relation to the presence or not of anti-CL-IgG antibodies.

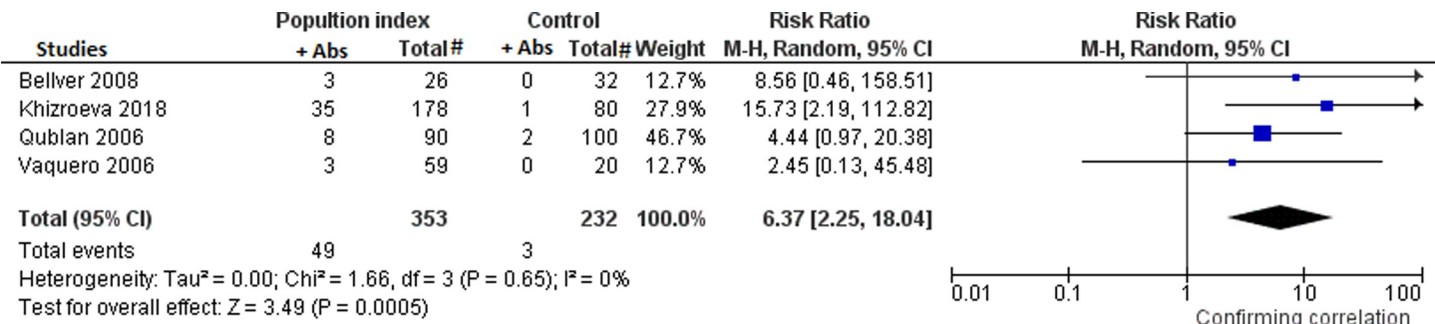

**Fig 6. Meta-analysis assessing the risk for implantation failure in relation to the presence or not of anti-PL antibodies in the studies in subgroup B between women with at least two implantation failures in IVF-ET (*population index*) *vs* women with at least one successful spontaneous pregnancy or unselected healthy fertile women with no history of IVF-ET (*control*).** Fig 6: meta-analysis assessing the risk for implantation failure in relation to the presence or not of LA antibodies.

rate, respectively, as compared to women experiencing one successful IVF-ET. In addition, in women experiencing at least two implantation failures in IVF-ET, presence of either anti-CL or LA or anti-$\beta_2$GPI or anti-PS antibodies is associated with a significant 13.92, 3.37, 15.04 and 164.58 RR for impaired implantation rate, respectively, as compared to women with at least one successful spontaneous pregnancy or unselected healthy fertile women with no history of IVF-ET. Most importantly, presence of either anti-$\beta_2$GPI or the rarely evaluated anti-PS antibodies in 13% and 41%, respectively, of all women experiencing at least two implantation failures in IVF-ET and in virtually none of control subjects suggests that these may be very accurate biomarkers (more accurate than the more frequently measured anti-PL antibodies) and urges further evaluation of their potential clinical use in infertility, as well as in APS in general. This is the first meta-analysis evaluating presence of any type of anti-PL antibodies in women experiencing implantation failure in IVF-ET. The results from the 17 selected studies were consistent, of strong methodological quality and support an association between recurrent implantation failure in IVF-ET and presence of anti-PL antibodies. This meta-analysis looked into multiple anti-PL antibodies, including newer markers (i.e. anti-PS antibodies) aiming at reporting a quantitative result based on the homogeneity and similarity in the findings of the included studies. Both in the retrospective and prospective studies included in the present meta-analysis the compared population indices and control groups are well defined. Thus, the accuracy and reliability of the extracted results cannot be limited in either case. The main limitation of this meta-analysis is the lack of relevant prospective, controlled studies and the resulting small number of included studies. Despite the small number of included studies, the conclusions of the present meta-analysis can be considered scientifically valid due to the

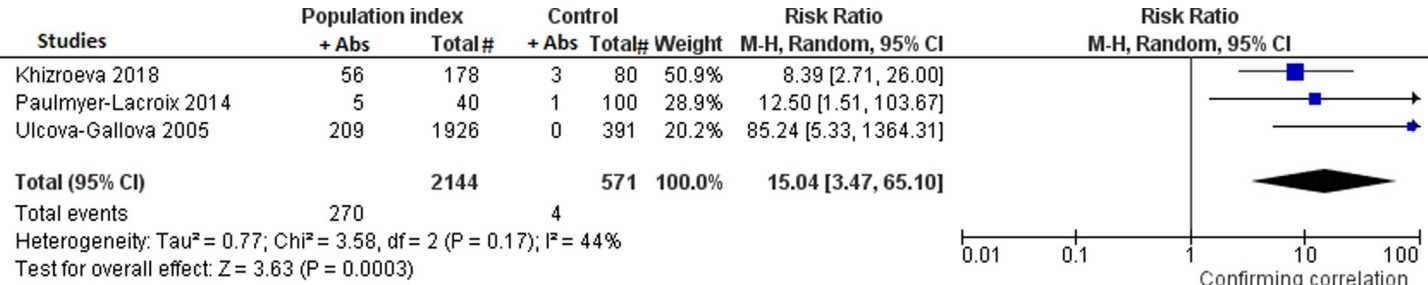

**Fig 7. Meta-analysis assessing the risk for implantation failure in relation to the presence or not of anti-PL antibodies in the studies in subgroup B between women with at least two implantation failures in IVF-ET (*population index*) *vs* women with at least one successful spontaneous pregnancy or unselected healthy fertile women with no history of IVF-ET (*control*).** Fig 7: meta-analysis assessing the risk for implantation failure in relation to the presence or not of anti- $\beta_2$GPI antibodies.

**Fig 8. Meta-analysis assessing the risk for implantation failure in relation to the presence or not of anti-PL antibodies in the studies in subgroup B between women with at least two implantation failures in IVF-ET (*population index*) *vs* women with at least one successful spontaneous pregnancy or unselected healthy fertile women with no history of IVF-ET (*control*).** Fig 8: meta-analysis assessing the risk for implantation failure in relation to the presence or not of anti-PS antibodies. +Abs: positive antibodies, total #: total number of participants.

*lege artis* mathematical approach followed in this meta-analysis and the homogeneity of the included studies.

Except the above mentioned anti-PL antibodies, which were evaluated by meta-analysis, this systematic review pointed towards other types of anti-PL antibodies, which seem to impair implantation and consequently fertility. Presence of specific types of anti-PL antibodies (i.e. anti-PS, anti-PC, anti-PI and anti-PA antibodies) was significantly increased in women with at least two implantation failures in IVF-ET and was associated with increased implantation failure rates, although specific types of anti-PL antibodies are in debate in the literature. Of note, these antibodies were not present in the respective control groups (Tables 1 and 2). The prevalence of these specific types of anti-PL antibodies reaches 40.4% among women with at least two implantation failures in IVF-ET compared to controls (range of prevalence among population index 2.4%-40.4%). Prevalence of anti-PE and anti-PG antibodies was also increased in women with at least two implantation failures in IVF-ET (range among population index 7.1%-19.6%) compared to their respective controls (range among controls: 0–4.8%) (Tables 1 and 2). It is noteworthy that the aforementioned types of anti-PL (anti-PS, anti-PC, anti-PI and anti-PA anti-PE and anti-PG antibodies) are rarely evaluated in clinical practice. One might suspect that they might be positive where other more often evaluated types of anti-PL such as anti-CL, LA or anti-$\beta_2$GPI antibodies are negative. Of note, in this meta-analysis, were included not only studies reporting anti-PL antibodies included in the revised Sapporo criteria (LA, anti-CL, anti-$\beta_2$GP I) but also anti-PL antibodies (anti-PS, anti-PC, anti-PE, anti-PI, anti-PG, anti-PA) which have gained importance for APS diagnosis in recent literature. Thus, it should be investigated whether these antibodies, not routinely measured, are involved as well in pathophysiologic aspects of implantation. Thus, questions emerge about their involvement in the underlying pathophysiological mechanisms in implantation.

The possible association of anti-PL antibodies with female infertility has been suggested since 1980s. Women with APS and women with anti-PL antibodies may present with impaired ovarian follicles reserve and more frequently with premature ovarian failure [29–31]. A retrospective study suggested that the presence of APS or just anti-PL antibodies are frequently encountered acquired risk factors for recurrent pregnancy loss and that they are associated with increased risk for ischemic placental dysfunction, such as fetal growth restriction, preeclampsia, premature birth and intrauterine death [32]. Till now, little is known about the biological mechanisms involved in the recurrent ET failures observed in presence of anti-PL antibodies as well as in APS. Anti-PS antibodies bind to human trophoblast in a dose-dependent way affecting thus, trophoblast invasiveness and differentiation of cytotrophoblast into a syncytium. It is also shown that, specifically anti-PS and not anti-CL antibodies are responsible for the decrease of hCG production by the placenta [33]. *In vitro* and *in vivo* studies, have suggested that anti-PL antibodies might affect negatively conception, implantation as well as early

and recurrent miscarriages. They can impair spontaneous as well as IVF-ET implantation as they are directed against negatively charged phospholipids located in the blood vessels of the uterine mucous membrane, or on the surface of oocytes, or they can affect the early embryo at the initial implantation process [32]. The latter and the maintenance of pregnancy in its early stages could be affected by inhibition of prostaglandin synthesis caused by anti-PL antibodies [32]. The latter affect maternal blood vessels, decidua and trophoblasts. They have been suggested to target tissue plasminogen activator, plasmin, annexin A2 and thrombin [34]. Prothrombin and $\beta_2$GPI mediate binding of anti-PL antibodies to target cells such as endothelial cells, monocytes, platelets and trophoblasts leading to thrombosis of placental vessels and fetal loss [35, 36]. Anti-PL antibodies and particularly $\beta_2$GPI-mediated anti-PL antibodies bind to trophoblast monolayers and can induce direct cellular injury, inhibition of proliferation and syncytia formation, apoptosis, and defective invasiveness [35, 36]. In mice, $\beta_2$GPI is essential for a successful pregnancy and for optimal placental development [12]. *In vitro* studies indicate that anti-CL antibodies inhibit trophoblast proliferation possibly by a prostacyclin-thromboxane A2 imbalance [37, 38]. Patients' anti-PS antibodies when co-cultured with rats embryos delay the development of rat yolk sacs [35]. These antibodies have been shown to affect negatively implantation in rats while they bind to human trophoblast in a dose dependent way affecting thus, trophoblast invasiveness and differentiation of cytotrophoblast into a syncytium [33]. To our knowledge, the presence of anti-PC, anti-PE, anti-PI, anti-PG, or anti-PA antibodies has not as yet been associated directly with implantation failure.

In summary, this meta-analysis has shown that presence of any type of anti-PL antibodies is associated with impaired implantation among women experiencing at least two implantation failures in IVF-ET and not suffering from APS compared to either women experiencing one successful IVF-ET or women with at least one successful spontaneous pregnancy or unselected healthy fertile women with no history of IVF-ET. Presence of anti-$\beta_2$GPI and anti-PS antibodies suggests an excessive risk. Importantly, types of anti-PL, not frequently measured in daily medical practice (anti-PS, anti-PC, anti-PE, anti-PI, anti-PA and anti-PG antibodies), seem to be stronger predictors of implantation failure in IVF-ET. In women not suffering from APS, presence of antibodies (lupus anticoagulant, anti-cardiolipin antibodies and anti-beta$_2$-glycoprotein I antibodies) included in the definition of APS, seem to be strongly associated with implantation failure in IVF-ET. In guidelines, it is suggested to evaluate anti-PL antibodies in women suffering from recurrent miscarriages [5]. Similarly, in women presenting multiple implantation failures in IVF-ET, it could be suggested to measure these antibodies, in order to investigate causality and, eventually, suggest treatment when additional studies will be available.

Well designed randomized controlled trials are needed in order to understand the impact of different types of anti-PL on implantation and consequently on infertility, before using them in everyday clinical management of infertile women with recurrent ET failures. Moreover, cost-effectiveness studies should be conducted to evaluate benefits and costs of this approach. Well designed interventional studies might confirm presence of anti-PL antibodies as predictive markers of implantation failure, but also target them pharmacologically in women suffering from infertility or subfertility. The present meta-analysis highlights the importance of the presence of anti-$\beta_2$GPI and anti-PS antibodies regarding the risk for implantation failure. Thus, it would be useful to insist on measuring them in cases of infertility at least concurrently with the more frequently measured aPL antibodies.

## Supporting information

**S1 Appendix.**
(DOCX)

**S1 Checklist. PRISMA 2009 checklist.**
(DOC)

## Author Contributions

**Conceptualization:** Eirini Papadimitriou, Georgios Boutzios.

**Data curation:** Eirini Papadimitriou.

**Formal analysis:** Eirini Papadimitriou.

**Methodology:** Alexander G. Mathioudakis.

**Project administration:** Eirini Papadimitriou, Georgios Boutzios.

**Software:** Eirini Papadimitriou, Alexander G. Mathioudakis.

**Supervision:** George Mastorakos.

**Writing – original draft:** Eirini Papadimitriou.

**Writing – review & editing:** Eirini Papadimitriou, Nikos F. Vlahos, Panayiotis Vlachoyianno-poulos, George Mastorakos.

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
