## [Decision Letter · Decision Letter 0]

11 May 2021

PONE-D-21-07054

Presence of antiphospholipid antibodies is associated with increased implantation failure following in vitro fertilization technique and embryo transfer: A systematic review and meta-analysis.

PLOS ONE

Dear Dr. Mastorakos,

Thank you for submitting your manuscript to PLOS ONE. After careful consideration, we feel that it has merit but does not fully meet PLOS ONE’s publication criteria as it currently stands. Therefore, we invite you to submit a revised version of the manuscript that addresses the points raised during the review process.

The review deals with an interesting topic and adequately follows the reference methodological checklist.

In addition to the referees' comments which deserve careful consideration, I have the following observations:

- I suggest elaborating figure 1 in detail according to the Prisma 2020 version.

- The tables are not immediately understandable and have very long titles. I suggest changing them with a shorter title, more immediate and clear content and adding any footnotes.

- in forest plots it is necessary to modify the "success IVF" outcome with a more informative title (pregnancy rate? Delivery rate? ...)

- when the I2 index is quite high, for example> 30%, it may be advisable to also provide the results using the random effect model, or to discuss why only the fixed effect model was used.

We look forward to receiving your revised manuscript.

Kind regards,

Alessio Paffoni, PhD

Academic Editor

PLOS ONE

Journal Requirements:

"Alexander G. Mathioudakis is supported by the National  Institute of Health Research

 Manchester Biomedical Research Centre (NIHR Manchester BRC)."

"he authors received no specific funding for this work."

Reviewers' comments:

Reviewer's Responses to Questions

**Comments to the Author**

1. Is the manuscript technically sound, and do the data support the conclusions?

Reviewer #1: Yes

Reviewer #2: Yes

2. Has the statistical analysis been performed appropriately and rigorously? 

Reviewer #1: Yes

Reviewer #2: Yes

3. Have the authors made all data underlying the findings in their manuscript fully available?

Reviewer #1: Yes

Reviewer #2: Yes

4. Is the manuscript presented in an intelligible fashion and written in standard English?

Reviewer #1: Yes

Reviewer #2: Yes

5. Review Comments to the Author

Reviewer #1: Title:

- Concise and sound

Abstract:

- Purpose: Can you please make it more clear that you are comparing aPL between patients having IVF-ET failures and controls. No need to mention implantation rate.

- Methods: Clear

- Results: Clear

- Conclusion: Please highlight the important finding that aB2GPI and aPS are the most associated with implantation failures.

Keywords:

- Please decrease the number, no need to mention the types of aPL

Introduction:

- Line 57: Reference 4 should be replaced with a specific ObGyn reference

- Line 62: Please add a reference

- Line 63: Replace the word biochemical with serological

- Line 67 to 69: I would suggest to remove because these non-criteria aPL are gaining attention in the current research

- Line 70: Why are you defining aPL again, this should happen previously

- Line 75: Please add a reference

- Lines 75- 77: Please add little bit more data about the pathophysiology, although you have elaborated precisely in the discussion part

Materials and Methods:

- Line 92: Screening by one author is a limitation. Please explain.

- Line 100: The past medical history needs elaboration. Any woman with previous diagnosis of APS or thrombosis?

- Line 123: Here, you mentioned that you will be assessing non-criteria aPL which means that they have clinical significance contrary to what you have preciously mentioned in the introduction.

- Line 149: Can you please submit a clearer version of figure 1, I can barely see it.

- Line 156: Can you please elaborate about the prospective nature of the studies (Trial, cohort, etc...)

- Line 160: After how many days did the studies assess for hCG?

- Line 181: I would suggest mentioning the correct references for the cut-offs and manufacturer.

Results:

- Please explain what you mean by population index, which will make it easier for the general reader.

- Lines 235-239: Here, you compare the isotypes of aCL. What is intention? Later on, you completely neglect the significant findings.

Discussion:

- Lines 279- 282: Please make shorter.

- Lines 282-287: How did the data differ between retrospective and prospective studies? Could be this a limitation?

- Lines 294-301: Here, you are discussing non-criteria aPL. Were these confirmed in any of the studies that you have used? Some non-criteria aPL are still considered acute-phase reactants.

- Line 304: Would suggest using better references. For example, Rodrigues VO, Soligo AGES, Pannain GD. Antiphospholipid Antibody Syndrome and Infertility. Rev Bras Ginecol Obstet. 2019 Oct;41(10):621-627. English. doi: 10.1055/s-0039-1697982. Epub 2019 Oct 28. PMID: 31658490. and El Hasbani G, Khamashta M, Uthman I. Antiphospholipid syndrome and infertility. Lupus. 2020 Feb;29(2):105-117. doi: 10.1177/0961203319893763. Epub 2019 Dec 12. PMID: 31829084.

- Lines 304- 310: We already know that APS can cause early miscarriage. This is out of the scope of the systematic review.

- Line 319: Reference please

- Line 324: Reference please

- Lines 327- 332: What about other non-criteria aPL? Do we have preliminary data on their relationship with implantation failure?

- The discussion would benefit more from highlighting the important finding of the role of aB2GPI and aPS in implantation failure.

Figures:

- Would suggest making them more clear to the general reader. What do you mean by panel A, B, etc... Please elaborate and guide us through every figure.

References:

References 21, 25, 30, and 38 need proper citations. There are errors in these references.

Reviewer #2: This is an interesting study and the authors have collected a unique dataset. The paper is generally well written and structured. May be only a few remarks:

1) 62 line - "Clinical manifestations of APS include fertility difficulties and pregnancy complications such as repeated miscarriages".

- Please add thrombosis as a one of the main manifestations of APS (besides obstetric complications)

2) 64-65 line - "When circulating anti-PL antibodies are positive on two occasions six weeks to six months apart, the diagnosis of APS is confirmed".

- In fact, there are no data to validate this interval but updated APS criteria increased the interval from 6 to 12 weeks. So, officially this interval consists of 12 weeks.

Overall, this is a clear, concise, and well-written manuscript that deserved to be accepted and published.

6. PLOS authors have the option to publish the peer review history of their article (what does this mean?). If published, this will include your full peer review and any attached files.

Reviewer #1: No

Reviewer #2: **Yes: **Khizroeva

---

## [Author Response · Author response to Decision Letter 0]

28 Oct 2021

George Mastorakos, MD, DSc

Professor of Endocrinology

University of Athens

Greece

3, Neofytou Vamva st

10674 Athens

Greece

Fax: 0030-210-3636229

e-mail: mastorakg@gmail.com

 Alessio Paffoni, PhD

Academic Editor

PLOS ONE

July 30th , 2021,

Dear Dr Alessio Paffoni,

We have received your revision letter of our paper entitled: "Presence of antiphospholipid antibodies is associated with increased implantation failure following in vitro fertilization technique and embryo transfer: A systematic review and meta-analysis." (PONE-D-21-07054). We thank you and the reviewers for your valuable time and appreciate the comments to which we respond as follows. 

Editor’s comments: 

Comment 1. I suggest elaborating figure 1 in detail according to the Prisma 2020 version.

Response to comment 1: We thank the editor for this useful suggestion. We have now elaborated figure 1 according to the Prisma 2020 version. 

Comment 2. The tables are not immediately understandable and have very long titles. I suggest changing them with a shorter title, more immediate and clear content and adding any footnotes.

Response to comment 2: We thank the editor for this remark. We have shortened the titles and added an explanatory footnote at each table. 

Comment 3. In forest plots it is necessary to modify the "success IVF" outcome with a more informative title (pregnancy rate? Delivery rate? ...) 

Response to comment 3: We thank the editor for the astute remark. We have now modified all the forest plots to show the risk for implantation failure in relation to the presence of antiphospholipid antibodies (+Abs). The term “success IVF” has been removed from the forest-plots and specific explanations are now included in the legends of the Figures.

Comment 4. When the I2 index is quite high, for example> 30%, it may be advisable to also provide the results using the random effect model, or to discuss why only the fixed effect model was used.

Response to comment 4: We thank the Editor for this comment. In line with guidance by Cochrane, in all provided results the random effect model has been employed. This is clearly stated in the Material and methods section (page 6, line 140-141): “… For data synthesis the random effect model was employed as significant clinical and methodological heterogeneity among the included studies was anticipated. …”. 

Reviewers’ comments

Reviewer #1

Title:

- Concise and sound

Response: We thank the reviewer for stating that the title of our manuscript was “concise and sound”. 

Abstract:

- Purpose: Can you please make it more clear that you are comparing aPL between patients having IVF-ET failures and controls. No need to mention implantation rate.

Response: We thank the reviewer for this remark. We have now rephrased the aim of the study to make it more clear that presence of aPL is compared between patients having IVF-ET failures and controls. It states now in the Abstract section (page 2, line 25-30): “…A systematic review and meta-analysis was conducted comparing the presence of anti-phospholipid (anti-PL) antibodies between women of reproductive age, without diagnosis of antiphospholipid syndrome, who experienced at least two implantation failures following in vitro fertilization and embryo transfer (IVF-ET), and either women who had a successful implantation after IVF-ET or women with at least one successful spontaneous pregnancy or unselected healthy fertile women with no history of IVF-ET. …”

- Methods: Clear

- Results: Clear

Response: We thank the reviewer for stating that the methods and the results in the abstract section were clear. 

- Conclusion: Please highlight the important finding that aB2GPI and aPS are the most associated with implantation failures.

Response: We thank the reviewer for this suggestion. We have now included the following comment in the Abstract section (page 2, line 44-45): “…The prevalence of antiphospholipid antibodies, particularly that of anti-beta2 glycoprotein-I and anti-phosphatidylserine antibodies, in women experiencing at least two implantation failures in IVF-ET…”. 

Keywords:

- Please decrease the number, no need to mention the types of aPL

Response: We have now decreased the number of keywords, as follows: “Antiphospholipid, anti-phosphatidylserine, anti-beta2 glycoprotein I, implantation, IVF”. 

Introduction:

- Line 57: Reference 4 should be replaced with a specific ObGyn reference

https://obgynkey.com/chapter-20-single-embryo-transfer/

Response: We thank the reviewer for this suggesion. As suggested reference 4 has now been replaced by: Martine Nijs, Single Embryo Transfer in Stewart, J. (Ed.). (2019). Subfertility, Reproductive Endocrinology and Assisted Reproduction. Cambridge: Cambridge University Press. doi:10.1017/9781316488294 

- Line 62: Please add a reference

Response: The following reference is now added: Miyakis S, Lockshin MD, Atsumi T, Tincani A, Vlachoyiannopoulos PG, Krilis SA et al. International consensus statement on an update of the classification criteria for definite antiphospholipid syndrome (APS). J. Thromb Haemost. 2006 Feb; 4(2):295-306. doi: 10.1111/j.1538-7836.2006.01753.x

- Line 63: Replace the word biochemical with serological

Response: We thank the reviewer for this suggestion. The word “biochemical” has now been replaced (page 3, line 65) with the word “serological”.

-Line 67 to 69: I would suggest to remove because these non-criteria aPL are gaining attention in the current research

Response: We thank the reviewer for this suggestion. Thus, the sentence (previously page 4, line 67-69): “… Other types of anti-PL antibodies have been employed in the past and still found in older literature although not included any more in the APS definition by international consensus committees. …” has been removed. 

- Line 70: Why are you defining aPL again, this should happen previously

Response: We thank the reviewer for this suggestion. Accordingly, we have moved the comment: “…Antiphospholipid antibodies represent a heterogeneous group of antibodies, which recognize various phospholipids, phospholipid-binding proteins, and phospholipid protein complexes. …” from previously line 70 to line 61-63. 

- Line 75: Please add a reference

Response: The following reference is now added: Carp HJ, Meroni PL, Shoenfeld Y. Autoantibodies as predictors doi: 10.1093/rheumatology/ken154

- Lines 75- 77: Please add little bit more data about the pathophysiology, although you have elaborated precisely in the discussion part

Response: We thank the reviewer for this suggestion. We have now added little more data to be more explanatory but not repetitive. It states now as follows (page 3, line 72-76): “…Antiphospholipid antibodies, especially anti-beta2 glycoprotein I (anti-β2GPI) antibodies, in pregnancy, appear to act directly on trophoblasts by activating pro-apoptotic and pro-inflammatory mechanisms6. At the same time, thrombosis of placental chorionic arteries and activation of the complement system intravascularly lead to the cell death of the trophoblast by decreasing trophoblast viability, syncytialization, and capacity for invasion. ....”.

Materials and Methods:

- Line 92: Screening by one author is a limitation. Please explain.

Response: We thank the reviewer for this remark. Literature was screened systematically by one author (EP) (as already mentioned in the manuscript) following the “Preferred reporting Items for Systematic Reviews and Meta-Analyses and detailed search strategy”which is available in the online supplement. Search strategy was validated by G.M. Wherever a discrepancy was raised by E.P., G.M. was consulted. This information is now added in the Materials and methods section (page 4, line 94-95): “…Search strategy was validated by GM. When EP raised a discrepancy, GM was consulted. …”. 

-Line 100: The past medical history needs elaboration. Any woman with previous diagnosis of APS or thrombosis?

Response: We thank the reviewer for raising this point. There was not any woman with previous diagnosis of APS or thrombosis. This it is stated in the inclusion criteria (page 5, line 108-109): “… All study populations should be consisted by healthy women of reproductive age not suffering from any known autoimmune, endocrine or infectious diseases. …” and it was followed accordingly. This piece of information is now included in the Material and methods section (page 4, line 100-101): “…past medical history of women included in each study was retrieved (no women suffered from APS or had a history of thrombosis)…”. 

- Line 123: Here, you mentioned that you will be assessing non-criteria aPL which means that they have clinical significance contrary to what you have previously mentioned in the introduction.

Response: We thank the reviewer for this astute remark. In the introduction section is stated: “According to revised Sapporo criteria, diagnosis of APS takes into account lupus anticoagulant (LA), anti-cardiolipin (anti-CL) antibodies or anti-β2glycoprotein I (anti-β2GP I) antibodies of either IgG or IgM isotype. Other types of anti-PL antibodies have been employed in the past and still found in older literature although not included any more in the APS definition by international consensus committees.” However in the present meta-analysis we aimed at comparing presence of aPL in women of reproductive age without diagnosis of antiphospholipid syndrome, who experienced implantation failure following in vitro fertilization and embryo transfer (IVF-ET) and in women who had a successful implantation after IVF-ET or in women with at least one successful spontaneous pregnancy or unselected healthy fertile women with no history of IVF-ET (please see response to Comment for abstract/purpose of the study of Reviewer #1). Thus, we opted to include comparisons of aPL included in the Sapporo criteria for APS diagnosis (anti-CL; LA; anti-β2GPI antibodies;) as well as of six more aPL (anti-PS; anti-PC; anti-PE; anti-PI; anti-PG; anti-PA antibodies) which gained importance in recent literature for APS diagnosis. This point is now clarified in the Material and methods section (page 5, line 119-124): “…Secondary outcomes extracted from the selected studies were presence or not of: anticardiolipin (anti-CL), lupus anticoagulant (LA) and anti-β2GPI antibodies (all three representing aPL included in the Sapporo criteria for APS diagnosis), as well as anti-phosphatidylserine (anti-PS), anti-phosphatidylcholine (anti-PC), anti-phosphatidylethanolamin (anti-PE), anti-phosphatidylinositol (anti-PI), anti-phosphatidylglycerol (anti-PG) and anti-phosphatidic acid (anti-PA) antibodies which have gained importance in recent literature for APS diagnosis. …”.

- Line 149: Can you please submit a clearer version of figure 1, I can barely see it.

Response: We thank the reviewer for this comment. Figure 1 was modified according the journal’s requirement and now a new version has been submitted.

- Line 156: Can you please elaborate about the prospective nature of the studies (Trial, cohort, etc...)

Response: We thank the reviewer for this remark. We have now clarified in Results section (page 7, line 155-156): ‘‘…(ten and seven studies were retrospective and prospective cohort studies, respectively)) …’’.

- Line 160: After how many days did the studies assess for hCG?

Response: Unfortunately, studies did not report after how many days hCG was determined. This is now clarified in the Results section (page 7, line 161-162): “…in none of the studies the timing of hCG measurement was reported …”.

- Line 181: I would suggest mentioning the correct references for the cut-offs and manufacturer.

Response: We thank the Reviewer for raising this point. 

This information is now included in the material and methods section (page 8, line 181-192): “…Diagnostic cut-offs for the antibodies were reported in: Qublan H. et al (anti-CL antibodies positive >10 IU/ml; qualitative positivity or negativity for LA)13; Vaquero E et al (anti-CL antibodies positive >20; qualitative positivity or negativity for LA)18; Bellver et al (anti-CL antibodies positive >20 gPL/ml or mPL/ml for IgG or IgM isotype, respectively; qualitative positivity or negativity for LA)21; Sanmarco M. et al [antibodies positive for anti-CL: IgG ≥20 GPLU; for anti-β2GPI IgG ≥10 B2GU; for aPE IgG ≥15 PEGU (GPLU, MPLU, B2GU and PEGU are arbitrary units for optical density)] 20 ; Geva E. et al (anti-CL antibodies positive >23 GPLU) 14.

In six studies positivity was based on optical density measurement exceeding the 99th or the 95th percentile of measurements established for each phospholipid in healthy individuals of reproductive age 11, 14, 17, 19, 22, 23. Six studies did not report diagnostic cut-offs 3, 9, 10, 15, 16, 24. Manufacturers of the assays are reported in all studies except two 13, 20. …”.

Manufacturers of the assays are reported in all studies except two (14, 21). This information is now reported in detail in the online Appendix in the characteristics of each study. 

Results:

- Please explain what you mean by population index, which will make it easier for the general reader.

Response: We thank the reviewer for this comment. We have now added in the Materials and Methods section the following clarification (page 4, line 93-94): “…The term population index refers to the total number of women who experienced at least two implantation failures after IVF-ET. …”.

- Lines 235-239: Here, you compare the isotypes of aCL. What is intention? Later on, you completely neglect the significant findings.

Response: We thank the reviewer for this remark. Three studies out of six in subgroup B point out two distinct aCL isotypes (aCL/Ig-G, aCL/Ig-M) in their reporting of aCL evaluation. For the sake of detailed reporting we decided to mention this information. However, meta-analysis was conducted only for the aCL/Ig-G isotype because, for this isotype, a non-significant heterogeneity among the selected studies was observed, whereas for the aCL/Ig-M isotype the heterogeneity observed among the selected studies was substantial. Thus, in line with the methodology and guidance of Cochrane handbook a meta-analysis was not meaningful for the aCL/Ig-M isotype. This information is now added in the Results section (page 11 line 245-250). “...Three studies out of six in subgroup B point out two distinct aCL isotypes (aCL/Ig-G, aCL/Ig-M) in their reporting of aCL evaluation. However, meta-analysis was conducted only for the aCL/Ig-G isotype because, for this isotype, a non-significant heterogeneity among the selected studies was observed, whereas for the aCL/Ig-M isotype the heterogeneity observed was substantial. Thus, in line with the methodology and guidance of Cochrane handbook, a meta-analysis was not meaningful for the aCL/Ig-M isotype. …”.

Discussion:

- Lines 279- 282: Please make shorter.

Response: We thank the reviewer for this recommendation. We have now made it shorter, as follows (page 12, line 295-297): “…This meta-analysis looked into multiple anti-PL antibodies, including newer markers (i.e. anti-PS antibodies) aiming at reporting a quantitative result based on the homogeneity and similarity in the findings of the included studies. …”.

-Lines 282-287: How did the data differ between retrospective and prospective studies? Could be this a limitation?

Response: We thank the reviewer for raising this point. The aim of this systematic review was to evaluate whether the presence of antiphospholipid antibodies has an impact on implantation rate. The retrospective studies compared the presence of antiphospholipid antibodies as a risk factor between women who experienced implantation failure after IVF-ET and controls, extracting this information from the medical history and previous laboratory. The prospective studies evaluate the presence of antiphospholipid antibodies between the women who experienced implantation failure after IVF-ET and controls, extracting this information after the enrollment of individuals it the study. The compared groups (population index and controls) are well defined in both cases. Thus, the accuracy and reliability of the extracted results cannot be limited in either case. This information is now added in the Discussion section (page 13-14 line 297-299): “...Both in the retrospective and prospective studies included in the present meta-analysis the compared population indices and control groups are well defined. Thus, the accuracy and reliability of the extracted results cannot be limited in either case. …”.

- Lines 294-301: Here, you are discussing non-criteria aPL. Were these confirmed in any of the studies that you have used? Some non-criteria aPL are still considered acute-phase reactants.

Response: We thank the reviewer for this comment. Indeed, as previously clarified (please see our response to the Comment for Line 123 of the Material and Methods section of this Reviewer), in this meta-analysis, were included not only studies which evaluated aPLs of the revised Sapporo criteria (LA, anti-CL, anti-β2GP I) but also types of anti-PL (anti-PS, anti-PC, anti-PE, anti-PI, anti-PG, anti-PA) antibodies which have gained importance in recent literature for APS diagnosis. Thus, it should be investigated whether these antibodies, not routinely measured, are involved as well in pathophysiologic aspects of implantation. To clarify this point, we rephrased as follows (page 13, line 317-322): “…Of note, in this meta-analysis, were included not only studies reporting anti-PL antibodies included in the revised Sapporo criteria (LA, anti-CL, anti-β2GP I) but also anti-PL antibodies (anti-PS, anti-PC, anti-PE, anti-PI, anti-PG, anti-PA) which have gained importance for APS diagnosis in recent literature. Thus, it should be investigated whether these antibodies, not routinely measured, are involved as well in pathophysiologic aspects of implantation. …”.

- Line 304: Would suggest using better references. For example, Rodrigues VO, Soligo AGES, Pannain GD. Antiphospholipid Antibody Syndrome and Infertility. Rev Bras Ginecol Obstet. 2019 Oct;41(10):621-627. English. doi: 10.1055/s-0039-1697982. Epub 2019 Oct 28. PMID: 31658490. and El Hasbani G, Khamashta M, Uthman I. Antiphospholipid syndrome and infertility. Lupus. 2020 Feb;29(2):105-117. doi: 10.1177/0961203319893763. Epub 2019 Dec 12. PMID: 31829084.

Response: We thank the reviewer for the suggestion. We have now replaced the references.

- Lines 304- 310: We already know that APS can cause early miscarriage. This is out of the scope of the systematic review.

Response: We thank the reviewer for suggesting this accurate improvement of our manuscript. We have now removed any reference to APS alone, although we kept references reporting data from presence of either APL or APS in the same study. Thus, the following sentence is now removed (previous page 14, line 307-310): “…In a 10-year follow-up of an observational, prospective study of 1000 patients (82% women) with primary or secondary APS, 127 (15.5%) women became pregnant (188 pregnancies) with 137 (72.9%) of them ending with healthy term neonate. The reported complications included early pregnancy losses (16.5% of all pregnancies) 34….”.

- Line 319: Reference please

Response: The following reference is now added: Nodler J et al. Elevated antiphospholipid antibody titers and adverse pregnancy outcomes: analysis of a population-based hospital dataset. BMC Pregnancy Childbirth. 2009 Mar 16;9:11 doi: 10.1186/1471-2393-9-11

- Line 324: Reference please

Response: The following references are now added: 

Marchetti et al. Obstetrical Antiphospholipid Syndrome: From the Pathogenesis to the Clinical and Therapeutic Implications. Clinical and Developmental Immunology Volume 2013 DOI: 10.1155/2013/159124

Arakawa M, Takakuwa K, Honda K, Tamura M, Kurabayashi T, Tanaka K. Suppressive effect of anticardiolipin antibody on the proliferation of human umbilical vein endothelial cells. Fertil Steril. 1999 Jun;71(6):1103-7. doi: 10.1016/s0015-0282(99)00117-x. PMID: 10360918

- Lines 327- 332: What about other non-criteria aPL? Do we have preliminary data on their relationship with implantation failure?

Response: In this section of the Discussion we report data which associate aPLs of the revised Sapporo criteria (LA, anti-CL, anti-β2GP I) with implantation failure . From the other types of anti-PL (anti-PS, anti-PC, anti-PE, anti-PI, anti-PG, anti-PA) antibodies, only anti-PS have been associated, to our knowledge, directly with implantation failure as reported in this section. To clarify this point we have now added the following (page 14, line 351-352): “…To our knowledge, the presence of anti-PC, anti-PE, anti-PI, anti-PG, or anti-PA antibodies has not as yet been associated directly with implantation failure. …”.

 - The discussion would benefit more from highlighting the important finding of the role of aB2GPI and aPS in implantation failure. 

Response: We thank the reviewer for stating that the findings of the role of anti-β2GPI and anti-PS are important. We have now highlighted these findings, as follows (page 15, line 371-374): “…The present meta-analysis highlights the importance of the presence of anti-β2GPI and anti-PS antibodies regarding the risk for implantation failure. Thus, it would be useful to insist on measuring them in cases of infertility at least concurrently with the more frequently measured aPL antibodies.…’’ 

 Figures:

- Would suggest making them more clear to the general reader. What do you mean by panel A, B, etc... Please elaborate and guide us through every figure.

Response: We thank the reviewer for the comment. Legends to Figures 2 and 3 have been rephrased to explain more accurately the content of the represented Panels in each Figure as follows: 

Figure 2: Meta-analyses assessing the risk for implantation failure in relation to the presence or not of anti-PL antibodies in the studies in subgroup A between women with at least two implantation failures in IVF-ET (population index) vs. women with one successful IVF-ET (control). Panel A: meta-analysis assessing the risk for implantation failure in relation to the presence or not of any type of anti-PL antibodies. Panel B: meta-analysis assessing the risk for implantation failure in relation to the presence or not of anti-CL antibodies. Panel C: meta-analysis assessing the risk for implantation failure in relation to the presence or not of LA antibodies. +Abs: positive antibodies

Figure 3: Meta-analysis assessing the risk for implantation failure in relation to the presence or not of anti-PL antibodies in the studies in subgroup B between women with at least two implantation failures in IVF-ET (population index) vs women with at least one successful spontaneous pregnancy or unselected healthy fertile women with no history of IVF-ET (control). Panel A: meta-analysis assessing the risk for implantation failure in relation to the presence or not of anti-CL-IgG antibodies. Panel B: meta-analysis assessing the risk for implantation failure in relation to the presence or not of LA antibodies. Panel C: meta-analysis assessing the risk for implantation failure in relation to the presence or not of anti- β2GPI antibodies. Panel D: meta-analysis assessing the risk for implantation failure in relation to the presence or not of anti-PS antibodies. +Abs: positive antibodies

References:

-References 21, 25, 30, and 38 need proper citations. There are errors in these references.

Response: The previous references 21, 25 and 38 have now been corrected. 

Reviewer #2: 

This is an interesting study and the authors have collected a unique dataset. The paper is generally well written and structured. May be only a few remarks:

We thank the reviewer for stating that: “…This is an interesting study and the authors have collected a unique dataset. The paper is generally well written and structured. …”. 

-1) 62 line - "Clinical manifestations of APS include fertility difficulties and pregnancy complications such as repeated miscarriages".

- Please add thrombosis as a one of the main manifestations of APS (besides obstetric complications)

Response: We thank the reviewer for the comment. We have now rephrased this point in the Introduction section (page 3, line 63-64): “…Clinical manifestations of APS include fertility problems and pregnancy complications (such as repeated miscarriages) as well as venous or arterial thrombosis. …”.

-2) 64-65 line "When circulating anti-PL antibodies are positive on two occasions six weeks to six months apart, the diagnosis of APS is confirmed".

- In fact, there are no data to validate this interval but updated APS criteria increased the interval from 6 to 12 weeks. So, officially this interval consists of 12 weeks.

Response: We thank the reviewer for this remark. We have now clarified this in the Iintroduction section (page 3, line 66-67): “…When circulating anti-PL antibodies are positive at initial diagnosis, testing should be repeated at least 12 weeks later to confirm diagnosis of APS. …”. 

We have responded to all of the Reviewers’ comments. We believe, and hope that you will agree as well, that we have addressed satisfactorily all points raised so that our paper will be found adequate for publication in your prestigious Journal.

With best regards

George Mastorakos, MD, DSc

Professor of Endocrinology

University of Athens Medical School

e-mail: mastorakg@gmail.com

(On behalf of the authors)

---

## [Editor Report · Decision Letter 1]

17 Nov 2021

Presence of antiphospholipid antibodies is associated with increased implantation failure following in vitro fertilization technique and embryo transfer: A systematic review and meta-analysis.

PONE-D-21-07054R1

Dear Dr. Mastorakos,

We’re pleased to inform you that your manuscript has been judged scientifically suitable for publication and will be formally accepted for publication once it meets all outstanding technical requirements.

Kind regards,

Alessio Paffoni, PhD

Academic Editor

PLOS ONE

Additional Editor Comments (optional):

My opinion is that the authors responded comprehensively to the referees' comments. The manuscript in its present form is satisfactory and meets the publication criteria
---

## [Editor Report · Acceptance letter]

15 Dec 2021

PONE-D-21-07054R1 

Presence of antiphospholipid antibodies is associated with increased implantation failure following *in vitro* fertilization technique and embryo transfer: A systematic review and meta-analysis. 

Dear Dr. Mastorakos:

I'm pleased to inform you that your manuscript has been deemed suitable for publication in PLOS ONE. Congratulations! Your manuscript is now with our production department. 

Kind regards, 

on behalf of

Dr. Alessio Paffoni 

Academic Editor

PLOS ONE